# Model-Based Condition Monitoring of the Sensors and Actuators of an Electric and Automated Vehicle

**DOI:** 10.3390/s23020887

**Published:** 2023-01-12

**Authors:** Shiqing Li, Michael Frey, Frank Gauterin

**Affiliations:** Institute of Vehicle System Technology, Karlsruhe Institute of Technology (KIT), Kaiserstraße 12, 76131 Karlsruhe, Germany

**Keywords:** electric vehicle, sensors, actuators, abnormal condition monitoring, model-based fault detection, symptom generation, fault diagnosis

## Abstract

Constant monitoring of driving conditions and observation of the surrounding area are essential for achieving reliable, high-quality autonomous driving. This requires more reliable sensors and actuators, as there is always the potential that sensors and actuators will fail under real-world conditions. The sensitive condition-monitoring methods of sensors and actuators should be used to improve the reliability of the sensors and actuators. They should be able to detect and isolate the abnormal situations of faulty sensors and actuators. In this paper, a developed model-based method for condition monitoring of the sensors and actuators in an electric vehicle is presented that can determine whether a sensor has a fault and further reconfigure the sensor signal, as well as detect the abnormal behavior of the actuators with the reconfigured sensor signals. Through the simulation data obtained by the vehicle model in complex road conditions, it is proved that the method is effective for the state detection of sensors and actuators.

## 1. Introduction

Autonomous cars have gained popularity in recent years, and several are presently being tested on public roads to improve their reliability in real-world situations. The goal of developing automated vehicle systems for today’s cars is to make drivers and passengers safer and more comfortable by taking over the tasks of the drivers. With the help of actuators, sensors, and data processing, mechatronic systems are being built into modern vehicles to improve their performance, range of functions, quality, and reliability [1,2]. The measurement of the dynamic characteristics of the vehicles and the observation of the surroundings, which is accomplished by several sensors, are crucial components in the realization of high-level autonomous driving. In autonomous vehicles, many different sensors will be equipped, such as the inertial measurement unit (IMU), steering-wheel angle sensor and wheel-speed sensor, etc. For example, the IMU plays an irreplaceable role in the lateral control of the car [3]. Although the current sensors are advanced, in real situations under long-term operation, there is always the possibility that the sensors may fail. In addition, these sensors are commercially accessible and employed in car-stability systems, but their price is an issue for manufacturers who are attempting to reduce costs [4]. In addition, the failed sensors will produce incorrect signals, which will then influence the control of the vehicle. On the other hand, steering and drive systems in modern electric vehicles are becoming increasingly automated by electric actuators, which simplifies driving tasks. However, for these automation benefits to be realized, each actuator must operate without fault. However, the additional electric actuators have increased the complexity of the vehicle because the causes of faults are becoming more complicated. In addition, with increasing requirements for high reliability and safety, actuator fault detection is becoming more and more important. To increase the reliability of the sensors and actuators, sensitive fault detection and diagnostic techniques should be used. In [5], fault tolerant designs have been suggested. These should have the capability to detect and isolate the presence and location of faults in the system, and then reconfigure the system architecture to compensate for those faults, which is called fault recovery. In this process, it is very important to determine the state of the system and the situation of the fault, which involves two steps—fault detection, and fault diagnosis. The common detection methods for complex systems are the model-based method, the signal-based method, and the knowledge-based method [6]. Among them, the model-based method is the most widely used method today, which is based on information from multiple sensors to build the analytical redundancy [2,3,4,7,8,9,10]. In the system, the sensors are related and will be executed in parallel to provide a group of features. Then, these features are compared with the system, forming residual values. Then, the residuals are entered into a fault-classification part to detect a fault and its location (fault diagnosis) [5]. For fault diagnostic, there are many different methods, such as neural networks, fuzzy logic, binary logic, evolutionary algorithms, and support vector machines [1,8,11].

In this paper, a model-based sensor and actuator condition-monitoring method is developed with the demonstration vehicle, which comes from the joint research project “SmartLoad” [12]. This method is developed in a simulation model with the use of CarMaker and Simulink and then validated under various scenarios to verify its reliability. Using this method, the faulty state of the six sensors, two steering actuators, and two driving actuators of the demonstrator can be detected, and the faulty state of the sensors can also be reconfigured.

## 2. Related Work

A systematic overview of fault detection and fault diagnosis is given in the research of Professor Isermann, who formalized the concept of the problem. The process of determining whether or not a fault has occurred in a monitored system is called fault detection. Using the dependencies that exist between various observable signals, it examines the processes, actuators, and sensors to identify any faults that may have occurred. Isolating the problem and locating it are additional duties that fall under this category. The extent of the fault is determined by the fault-identification process, whereas the location and kind of the fault are determined by the fault-isolation process. Fault diagnosis is a collective term that refers to both fault isolation and fault identification. The task of fault diagnosis consists of determining the type of fault, together with as many facts as possible about the fault [5].

### 2.1. Fault Detection

Fault detection is a subsection of control engineering, which is based on signal processing, pattern recognition, reliability theory, artificial intelligence, applied mathematics, etc. With the development and improvement of fault detection, the methods can be divided into the model-based method, the signal-based method, and the knowledge-based method [6].

Analytical redundancy is the foundation of model-based approaches [7]. The essence of this concept is the comparison of the actual outputs of the monitored system with the outputs obtained from an analytical mathematical model. It involves two stages: residual generation and residual evaluation. According to the different ways to generate residuals, the model-based method can be classified as state observers, parity equations, and parameter estimation [8].

The basic idea of state observers is to reconstruct or estimate the measurable variables of the system through the measured signals and mathematical model of the system, and detect and isolate faults using residual signals that are generated by comparing the actual measured value with the estimated value [8]. This method can detect faults in real time and does not require a continuous excitation signal [11]. In [2], the authors used a state estimator to create a bank of residuals and the adaptive threshold to identify the abnormal change in residual signals to detect the fault of the sensors and actuators of the vehicle. In [9], a designed observer was introduced that can monitor the health of lateral vehicle control systems, which includes a GPS system, a gyroscope, and an accelerometer. In [10], a sliding model was used in the design of a state observer for fault detection in a steer-by-wire vehicle. In [3], Kalman filters were applied to find the faulty sensors and approach good state estimation considering stochastic problems. In addition, a virtual sensor was designed in [4], which can be used to diagnose faults in a real sensor.

In the mathematical model of the system, parity equations define the dynamic, redundant connection between the input and output variables. When actual process behavior is compared to expected behavior, faults are found. Comparatively simpler to develop than observers are parity equations. Most of them are only useful in linear systems. Therefore, it is challenging to identify faults in nonlinear systems. Parity equations are sensitive to additive faults [13]. In [1], 12 models are used to yield the residuals, and then the calculated residuals are used for fault diagnosis. In [14], eight models were used to transform the measured wheel speeds, steering angle, and yaw rate into vehicle central axis information, and then a reference central axis speed was selected based on this information. Thus, the speeds for all wheel sides were estimated by the selected central speed, which were compared with measurements to identify faults.

The basic idea of parameter estimation is to diagnose faults in a process by detecting changes in parameters in the model. It requires that faults be reflected in parameters [13]. Reference parameters are compared with actual parameters, which are estimated by measurements using a mathematical model. In [1], parameters of an active suspension were estimated with the Discrete Square Root Filter in Information Form, which is based on the recursive least squares algorithm.

### 2.2. Fault Diagnosis

The objective of fault diagnosis is to determine the kind, the location, the size, and the time-varying behavior of the fault. It follows fault detection and includes fault isolation and identification. In general, there are two main types of diagnosis—classification methods and diagnostic reasoning methods. Recently, the field of classification methods, especially neural networks and fuzzy logic classification methods, has been developed. Likewise, rule-based reasoning methods are increasingly used in fault diagnosis [11]. If the fault–symptom relationships are not available in advance, a system that can be learned from experimental or simulated data is required. Then, with fault symptoms, the two fault-diagnosis methods can be applied—classification methods and inference methods [8].

If there is no clear causality between fault and symptom, classification methods can be trained for fault diagnosis, such as statistical and geometrical classification methods, neural networks, and fuzzy clustering. In [15], a fault-diagnosis method was introduced that is based on the combination of priory structural knowledge and measurement data to create a hierarchical diagnosis system. This method can be adapted to different motors. Because of a self-learning neuro-fuzzy scheme, this system is transparent and has increased robustness over traditional classification schemes.

The inference methods are applied with prior knowledge of symptom–cause relationships. For example, different sensor faults affect the symptoms in different ways, which means the change in symptom vectors is related to different sensor faults. Therefore, a fuzzy logic system based on the fault–symptom relationships can be used to diagnose the fault [1].

### 2.3. Contribution of This Work

It can be seen from the above literature review that there has been a lot of research on model-based fault detection, but in general, there are not so many studies about the complete condition monitoring of a whole vehicle, including sensors and electric actuators. Therefore, in this article, we will use the limited sensor information to perform model-based condition monitoring of the entire vehicle, including sensors and electric steering and driving actuators. We will first introduce how to use the model-based method to estimate the sensor value that we need, and then judge the state of the sensor by generating the residual and evaluating the residual. In this process, we will do a comprehensive calculation of the basic dynamic parameters of the vehicle, which is an improvement from other papers that only focus on lateral or longitudinal characteristics. If the sensor is in an abnormal state, we will use the estimated sensor value to replace the actual sensor value to perform a simple sensor recovery. After ensuring that the sensor value is not significantly abnormal, a state detection of the actuators will be performed. This method of first detecting the states of the sensors and then using the corrected sensor values to further detect the actuators is also not included in other related work. Since the symptoms generated by the model-based method are all transparent, in this paper, we mainly use the simple traditional logic gate relationship to deal with the fault symptoms for the purpose of simple abnormal state warnings of the sensors and actuators. The advantage of this method is that it is simple and effective, and requires less computation. This article is based on the master theses [16,17,18,19], which were tutored by the author of this article and continues to be developed and improved. The method discussed in this paper can be summarized in Figure 1.

## 3. Model-Based Sensor Condition Monitoring

Because this article is based on the demonstrator vehicle in the “SmartLoad” [12] project, we first briefly introduce the sensors contained in the demonstrator vehicle. The demonstrator vehicle has a steering-wheel angle sensor, a steering-wheel torque sensor, a force sensor for measuring the force on the tie rod, two wheel-speed sensors on the rear wheels and an IMU (Inertial Measurement Unit). Therefore, the following research will mainly focus on the condition monitoring of these six sensors. The main steps in the state detection of the sensors are designed to obtain the predicted values of the six sensors through different mathematical models, which were described in Section 3.1, Section 3.2, Section 3.3, Section 3.4, Section 3.5 and Section 3.6. To ensure the accuracy of the predicted model, the models were then verified in Section 3.7. After ensuring the accuracy of the models, the predicted values were compared with the measured values of the sensors to obtain the residuals, which was introduced in Section 3.8, and the residuals were further analyzed to obtain the state information of the sensors, which were described in Section 3.9.

### 3.1. Model of the Steering-Wheel Angle

#### 3.1.1. Model with 2-DOF System

A 2-DOF system combining the compliance, inertia and friction elements was used to model the steering system. As shown in Figure 2, there are two inertia elements in this model. One inertia element is for the steering-wheel to the torsion bar, and another inertia element is for the rack and pinion.

To describe the model, the following equations can be written according to Figure 2 [20]:(1)$$\left\{\begin{array}{c} I_{S+uC+IC+H}{\ddot{\delta }}_{S}+c_{D}{\dot{\delta }}_{S}-d_{TT}\left({\dot{S}}_{R}-{\dot{\delta }}_{S}i\right)i+k_{D}\delta_{S}-k_{TT}\left(S_{R}-\delta_{S}i\right)i=M_{S}-M_{FR,P}\\ m_{R}{\ddot{S}}_{R}+d_{R}{\dot{S}}_{R}+d_{TT}\left({\dot{S}}_{R}-{\dot{\delta }}_{S}i\right)+k_{R}S_{R}+k_{TT}\left(S_{R}-\delta_{S}i\right)=F_{PR}+F_{R}-F_{FR,R} \end{array}\right.$$
where
$I_{S+uC+IC+H}$ is the Inertia of the steering wheel and steering column,$m_{R}$ is the equivalent mass on the rack,$c_{D}$, $d_{TT}$, $d_{R}$ are the damping constant of steering wheel, rack and pinion connection and rack, respectively,$k_{D}$, $k_{TT}$, $k_{R}$ are the spring constant of steering wheel, rack and pinion connection and rack, respectively,$M_{S}$ is the steering-wheel torque and $M_{FR,P}$ is the concentrated frictional torque,$F_{R}$, $F_{P}R$ is the tie-rod forces on the rack and $F_{FR,R}$ is the concentrated frictional force,$\delta_{S}$, $S_{R}$, *i* are the steering-wheel angle, shift of the rack and the steering ratio, respectively.

From the original Equations (Equation 1), we can lump the friction torque and friction force into one variable. In addition, since there is no steering wheel on the real car, we can omit the part of the steering wheel. Therefore, the equations for the steering model can be described as follows:(2)$$\left\{\begin{array}{c} I_{S+uC+IC+H}{\ddot{\delta }}_{S}-d_{TT}\left({\dot{S}}_{R}-{\dot{\delta }}_{S}i\right)i-k_{TT}\left(S_{R}-\delta_{S}i\right)i=M_{S}-M_{FR,S}\\ m_{R}{\ddot{S}}_{R}+d_{TT}\left({\dot{S}}_{R}-{\dot{\delta }}_{S}i\right)+k_{TT}\left(S_{R}-\delta_{S}i\right)=F_{S}-k_{R}S_{R}-d_{R}{\dot{S}}_{R} \end{array}\right.$$
where
$M_{FR,S}$ is the total concentrated frictional torque,$F_{S}$ is total tie-rod forces on the rack

For the estimation of the steering-wheel angle, an unscented Kalman filter is applied. According to Equation (Equation 2), the state model can be represented as
(3)$$\underset{\dot{x}}{\underset{︸}{\left(\begin{array}{c} {\dot{S}}_{R}\\ {\dot{\delta }}_{S}\\ {\ddot{S}}_{R}\\ {\ddot{\delta }}_{S} \end{array}\right)}}=\underset{A}{\underset{︸}{\left(\begin{array}{cccc} 0 & 0 & 1 & 0\\ 0 & 0 & 0 & 1\\ -\frac{k_{TT}+k_{R}}{m_{R}} & \frac{k_{TT}i}{m_{R}} & -\frac{d_{TT}+d_{R}}{m_{R}} & \frac{d_{TT}i}{m_{R}}\\ \frac{k_{TT}i}{I_{S+uC+IC+H}} & -\frac{k_{TT}i^{2}}{I_{S+uC+IC+H}} & \frac{d_{TT}i}{I_{S+uC+IC+H}} & -\frac{d_{TT}i^{2}}{I_{S+uC+IC+H}} \end{array}\right)}}\underset{x}{\underset{︸}{\left(\begin{array}{c} S_{R}\\ \delta_{S}\\ {\dot{S}}_{R}\\ {\dot{\delta }}_{S} \end{array}\right)}}+\underset{B}{\underset{︸}{\left(\begin{array}{cc} 0 & 0\\ 0 & 0\\ 1 & 0\\ 0 & 1 \end{array}\right)}}\underset{u}{\underset{︸}{\left(\begin{array}{c} \frac{F_{S}}{m_{R}}\\ \frac{M_{S}-M_{FR,S}}{I_{S+uC+IC+H}} \end{array}\right)}}+\left(\begin{array}{c} w_{S}\\ w_{\delta }\\ w_{\dot{S}}\\ w_{\dot{\delta }} \end{array}\right)$$
where $w_{S}$, $w_{\delta }$, $w_{\dot{S}}$ and $w_{\dot{\delta }}$ are the process noise of the state model.

The observation model is defined as
(4)$$\underset{y}{\underset{︸}{\left(\begin{array}{c} S_{R}\\ \delta_{S} \end{array}\right)}}=\underset{C}{\underset{︸}{\left(\begin{array}{cccc} 1 & 0 & 0 & 0\\ 0 & 1 & 0 & 0 \end{array}\right)}}\underset{x}{\underset{︸}{\left(\begin{array}{c} S_{R}\\ \delta_{S}\\ {\dot{S}}_{R}\\ {\dot{\delta }}_{S} \end{array}\right)}}+\left(\begin{array}{c} n_{S}\\ n_{\delta } \end{array}\right)$$
where $n_{S}$, $n_{\delta }$ are the measurement noises of the sensor.

The mass $m_{R}$ contains not only the mass of the rack but also the equivalent mass of the suspension, which is related to the tires. In this work, the displacement of the rack $S_{R}$ is unavailable. Therefore, it will be calculated from the measurement of steering-wheel angle $\delta_{S}$ with a constant steering ratio between the steering wheel and rack *i*:(5)$$S_{R}=\delta_{S}\cdot i$$

After the estimation of steering-wheel angle ${\hat{\delta }}_{SKF}$, the residual of the steering-wheel angle is generated by comparing the estimated value with the measured signal:(6)$$r_{1}={\hat{\delta }}_{SKF}-\delta_{S}$$

#### 3.1.2. Model with Wheel-Steering Angle

In addition to calculating the steering angle with the Kalman filter mentioned above, the steering-wheel angle can also be calculated from the signals measured by the yaw rate from the IMU and the wheel-speed sensors using a simple kinematic formula. First, the front wheel-steering angles are calculated:(7)$$\begin{array}{c} {\hat{\delta }}_{fl}=\cot^{-1}[\cot\left(\tan^{-1}\left(\frac{\dot{\psi }\left(l_{f}+l_{r}\right)}{v_{rl}}\right)+\frac{t_{r}}{2(l_{f}+l_{r})}\right]\\ {\hat{\delta }}_{fr}=\cot^{-1}[\cot\left(\tan^{-1}\left(\frac{\dot{\psi }\left(l_{f}+l_{r}\right)}{v_{rr}}\right)-\frac{t_{r}}{2(l_{f}+l_{r})}\right] \end{array}$$
where
$\delta_{fl}$ is the left front wheel-steering angle,$\delta_{fr}$ is the right front wheel-steering angle,$l_{f}$ is the distance between center of gravity and front axle,$l_{r}$ is the distance between center of gravity and rear axle,$t_{r}$ is the track width,$\dot{\psi }$ is the yaw rate,$v_{rl}$ is the left rear wheel velocity,$v_{rr}$ is the right rear wheel velocity,

Then the average of the two wheel-steering angles as follows:(8)$${\hat{\delta }}_{f}=\frac{{\hat{\delta }}_{fl}+{\hat{\delta }}_{fr}}{2}$$
the steering wheel angle ${\hat{\delta }}_{S}$ is determined with the steering ratio $i_{steer}$:(9)$${\hat{\delta }}_{S}=\frac{{\hat{\delta }}_{f}}{i_{steer}}$$

After estimating the steering-wheel angle, the residue of the steering-wheel angle is formed by comparing the estimated value with the measured signal:(10)$$r_{2}={\hat{\delta }}_{S}-\delta_{S}$$

### 3.2. Model of the Steering-Wheel Torque

The 2 DOF model is used here again as the equation (Equation 2). In this way, it can be speculated from the equation that steering-wheel torque is approximately a linear function of the steering-wheel angle and the first-order, second-order derivative:(11)$${\hat{M}}_{S}=a_{1}\cdot \delta_{S}+a_{2}\cdot {\dot{\delta }}_{S}+a_{3}\cdot {\ddot{\delta }}_{S}+M_{FR,S}$$
where $a_{1},a_{2},a_{3}$ are the factors, which can be determined by simulated data. The approximated steering-wheel torque ${\hat{M}}_{S}$ is compared with the measured signal of steering-wheel torque sensor $M_{S}$, and the residual of the steering-wheel torque can be estimated:(12)$$r_{3}={\hat{M}}_{S}-M_{S}$$

### 3.3. Model of the Yaw Rate

The single-track model (or bicycle model) has a good performance estimate of side-slip angle and yaw rate at low lateral acceleration. The state model can be defined as follows:(13)$$\underset{\dot{x}}{\underset{︸}{\left[\begin{array}{c} \dot{\beta }\\ \ddot{\psi } \end{array}\right]}}=\underset{A}{\underset{︸}{\left[\begin{array}{cc} -\frac{c_{f}+c_{r}}{mv_{x}} & -\frac{c_{f}l_{f}-c_{r}l_{r}}{mv_{x}^{2}}-1\\ -\frac{c_{f}l_{f}-c_{r}l_{r}}{J_{z}} & -\frac{c_{r}l_{r}^{2}+c_{f}l_{f}^{2}}{J_{z}v_{x}} \end{array}\right]}}\underset{x}{\underset{︸}{\left[\begin{array}{c} \beta \\ \dot{\psi } \end{array}\right]}}+\underset{B}{\underset{︸}{\left[\begin{array}{c} \frac{c_{f}}{mv_{x}}\\ \frac{c_{f}l_{f}}{J_{z}} \end{array}\right]}}\underset{u}{\underset{︸}{\left[\begin{array}{c} \delta_{f} \end{array}\right]}}+\left[\begin{array}{c} w_{\beta }\\ w_{\dot{\psi }} \end{array}\right]$$
where
$w_{\beta }$ and $w_{\dot{\psi }}$ are the process noises,β is the side-slip angle,$\dot{\psi }$ is the yaw rate,$c_{f}$ and $c_{r}$ are the cornering stiffness of front and cornering stiffness of rear wheel,*m* is the vehicle mass,$v_{x}$ is the longitudinal velocity,$J_{z}$ is the vehicle yaw inertia.

The lateral acceleration and the yaw rate are used as measurement vectors, then the observation model can be defined as:(14)$$\underset{y}{\underset{︸}{\left[\begin{array}{c} a_{y}\\ \dot{\psi } \end{array}\right]}}=\underset{C}{\underset{︸}{\left[\begin{array}{cc} -\frac{c_{f}+c_{r}}{mv_{x}} & -\frac{c_{f}l_{f}-c_{r}l_{r}}{mv_{x}}\\ 0 & 1 \end{array}\right]}}\underset{x}{\underset{︸}{\left[\begin{array}{c} \beta \\ \dot{\psi } \end{array}\right]}}+\underset{D}{\underset{︸}{\left[\begin{array}{c} \frac{c_{f}}{m}\\ 0 \end{array}\right]}}\underset{u}{\underset{︸}{\left[\begin{array}{c} \delta_{f} \end{array}\right]}}+\left[\begin{array}{c} n_{a_{y}}\\ n_{\dot{\psi }} \end{array}\right]$$
where
$n_{a_{y}}$ and $n_{\dot{\psi }}$ are the measurement noises,$a_{y}$ is the lateral acceleration.

The estimated yaw rate is compared to the measured yaw rate to obtain the residual:(15)$$r_{4}=\hat{\dot{\psi }}-\dot{\psi }$$

### 3.4. Model of the Longitudinal and Lateral Acceleration

We used the two-track model to figure out the longitudinal and lateral accelerations. Figure 3 shows the most important vehicle parameters, movement variables, and forces that act on a two-track model.

The formula of the velocity $v_{SP}$ in the vehicle-fixed coordinate system is:(16)$$v_{SP}=\left[\begin{array}{c} v_{x}\\ v_{y}\\ v_{z} \end{array}\right]=\left[\begin{array}{c} v\cos(\beta )\\ v\sin(\beta )\\ 0 \end{array}\right]$$

Then the center of gravity acceleration $a_{SP}$ can be calculated:(17)$$a_{SP}=\left[\begin{array}{c} a_{x}\\ a_{y}\\ a_{z} \end{array}\right]={\dot{v}}_{SP}+\omega \times v_{SP}=\left[\begin{array}{c} -v\sin\left(\beta \right)\dot{\beta }\\ v\cos\left(\beta \right)\dot{\beta }\\ 0 \end{array}\right]+\left[\begin{array}{c} 0\\ 0\\ \dot{\psi } \end{array}\right]\times \left[\begin{array}{c} v\cos(\beta )\\ v\sin(\beta )\\ 0 \end{array}\right]=\left[\begin{array}{c} -v\left(\dot{\psi }+\dot{\beta }\right)\sin\left(\beta \right)\\ v\left(\dot{\psi }+\dot{\beta }\right)\cos\left(\beta \right)\\ 0 \end{array}\right]$$

The lateral acceleration $a_{y}$ of the vehicle can be determined from the two-track model:(18)$$a_{y}=v_{x}\left(\dot{\psi }+\dot{\beta }\right)$$
where $\dot{\psi }$ is available from the sensor signal and $\dot{\beta }$ will be calculated as the model in Section 3.3.

The longitudinal acceleration expression of the vehicle can be obtained according to vehicle dynamics:(19)$$a_{x}={\dot{v}}_{x}-v_{x}\tan\left(\beta \right)\dot{\psi }$$

Thus, the residual of accelerations can be obtained by comparing the generated value with the sensor data:(20)$$r_{5}={\hat{a}}_{x}-a_{x}$$
(21)$$r_{6}={\hat{a}}_{y}-a_{y}$$

### 3.5. Model of the Rear Wheel Speed

#### 3.5.1. Model with the Average of the Two Central Axis Speed

The left and right rear wheel speeds are estimated through the kinematic model, which is also used in [14]. The basic idea is to convert measured wheel speeds into vehicle central axis information with steering-wheel angle $\delta_{s}$ (front wheel-steering angle $\delta_{f}$) and yaw rate $\dot{\psi }$. Here the central axis speed refers to the speed of the center of gravity of the car, because there are two calculation methods for the wheel-steering angle, so two kinds of speeds can also be obtained, and then the average of the two speeds is taken as the final speed of the center of gravity of the car. The average of the central axis velocity is then used to estimate the speed for both rear wheels, which are compared with measured values to yield the residuals.

There are two ways to calculate the wheel-steering angle. The wheel-steering angle can be estimated from the steering-wheel angle with the equation (Equation 9). In addition, it can also be estimated using the yaw rate, which was introduced in Section 3.1.2 with Equation (Equation 7). We can also obtain two corresponding values for side-slip angle β and curvature *C* from Equations (Equation 22) and (Equation 23).
(22)$$\beta =\tan^{-1}\left(\frac{l_{r}\tan\left(\delta_{f}\right)}{l_{f}+l_{r}}\right)$$
(23)$$C=\frac{1}{R}=\frac{\cos\left(\beta \right)\tan\left(\delta_{f}\right)}{l_{f}+l_{r}}$$

We assign the side-slip angle β and curvature *C* to Equations (Equation 24)–(Equation 27) using individual wheel speeds, and then we could obtain four central axis speeds $v_{crl\delta }$, $v_{crr\delta }$, $v_{crl\dot{\psi }}$, $v_{crr\dot{\psi }}$.
(24)$$v_{crl\delta }=\frac{v_{rl}}{\sqrt{\cos^{2}\left(\beta \right)+{\left(\frac{t_{r}}{2}\right)}^{2}\cdot C^{2}-t_{r}\cdot C\cdot \cos\left(\beta \right)}}$$
(25)$$v_{crr\delta }=\frac{v_{rr}}{\sqrt{\cos^{2}\left(\beta \right)+{\left(\frac{t_{r}}{2}\right)}^{2}\cdot C^{2}+t_{r}\cdot C\cdot \cos\left(\beta \right)}}$$
(26)$$v_{crl\dot{\psi }}=\frac{v_{rl}}{\sqrt{\cos^{2}\left(\beta \right)+{\left(\frac{t_{r}}{2}\right)}^{2}\cdot C^{2}-t_{r}\cdot C\cdot \cos\left(\beta \right)}}$$
(27)$$v_{crr\dot{\psi }}=\frac{v_{rr}}{\sqrt{\cos^{2}\left(\beta \right)+{\left(\frac{t_{r}}{2}\right)}^{2}\cdot C^{2}-t_{r}\cdot C\cdot \cos\left(\beta \right)}}$$

Calculate the average of two central axis speeds as:(28)$$v_{c\_\delta }=\frac{v_{crl\delta }+v_{crr\delta }}{2}$$
(29)$$v_{c\_\dot{\psi }}=\frac{v_{crl\dot{\psi }}+v_{crr\dot{\psi }}}{2}$$

Then the average of the central axis speed is employed to estimate the speed of each rear wheel with Equations (Equation 30)–(Equation 33).
(30)$${\hat{v}}_{rl\_\delta }=v_{c\_\delta }\sqrt{\cos^{2}\left(\beta \right)+{\left(\frac{t_{r}}{2}\right)}^{2}\cdot C^{2}-t_{r}\cdot C\cdot \cos\left(\beta \right)}$$
(31)$${\hat{v}}_{rr\_\delta }=v_{c\_\delta }\sqrt{\cos^{2}\left(\beta \right)+{\left(\frac{t_{r}}{2}\right)}^{2}\cdot C^{2}+t_{r}\cdot C\cdot \cos\left(\beta \right)}$$
(32)$${\hat{v}}_{rl\_\dot{\psi }}=v_{c\_\dot{\psi }}\sqrt{\cos^{2}\left(\beta \right)+{\left(\frac{t_{r}}{2}\right)}^{2}\cdot C^{2}-t_{r}\cdot C\cdot \cos\left(\beta \right)}$$
(33)$${\hat{v}}_{rr\_\dot{\psi }}=v_{c\_\dot{\psi }}\sqrt{\cos^{2}\left(\beta \right)+{\left(\frac{t_{r}}{2}\right)}^{2}\cdot C^{2}+t_{r}\cdot C\cdot \cos\left(\beta \right)}$$

The speeds with index δ indicate that the side-slip angle and curvature are calculated with the steering-wheel angle. In addition, the speed with index $\dot{\psi }$ indicates that the side-slip angle and curvature are calculated with the yaw rate.

#### 3.5.2. Model with Kalman Filter for Longitudinal Speed

The estimation of the left and right rear wheel speeds will be carried out in two steps. First, a two-stage adaptive Kalman filter, which was designed first in [21], is used to estimate the longitudinal speed of the vehicle, and then this speed is employed to calculate the left and right rear wheel speeds. The structure of the process is shown in Figure 4.

Stage I of the two-stage adaptive Kalman filter consists of two Kalman filters. The Kalman filter I is responsible for obtaining the derivative of the longitudinal acceleration ${\hat{a}}_{v_{x}}$ from the longitudinal acceleration signal $a_{x}$, the measured yaw rate $\dot{\psi }$, and the estimated side-slip angle $\hat{\beta }$ from the model in Section 3.3. The Kalman filter II deals with calculating the wheel circumferential speeds ${\hat{v}}_{xw<pos>}$ for avoiding the delay of signal, and, in this calculation, the yaw rate is taken into account. Moreover, the Kalman filter II attenuates the noise and delay from the signals for smooth output signals.

Stage II consists of the Kalman filter III, which is responsible for estimating the longitudinal speed.

As introduced in Section 3.5.1, we could now replace the average central axis speed in (Equation 30)–(Equation 33) with the Kalman filter longitudinal speed. In addition, we could also obtain the four-wheel speed: ${\hat{v}}_{rl\_KF\_\delta }$, ${\hat{v}}_{rr\_KF\_\delta }$, ${\hat{v}}_{rl\_KF\_\dot{\psi }}$, ${\hat{v}}_{rr\_KF\_\dot{\psi }}$.

Then we could obtain a total of eight residuals using the above methods:(34)$$r_{7}={\hat{v}}_{rl\_\dot{\psi }}-v_{rl}$$
(35)$$r_{8}={\hat{v}}_{rl\_KF\_\dot{\psi }}-v_{rl}$$
(36)$$r_{9}={\hat{v}}_{rl\_\delta }-v_{rl}$$
(37)$$r_{10}={\hat{v}}_{rl\_KF\_\delta }-v_{rl}$$
(38)$$r_{11}={\hat{v}}_{rr\_\dot{\psi }}-v_{rr}$$
(39)$$r_{12}={\hat{v}}_{rr\_KF\_\dot{\psi }}-v_{rr}$$
(40)$$r_{13}={\hat{v}}_{rr\_\delta }-v_{rr}$$
(41)$$r_{14}={\hat{v}}_{rr\_KF\_\delta }-v_{rr}$$

In addition, since we also estimate the velocity and acceleration with Kalman filter, we can also obtain another two residuals:(42)$$r_{15}={\hat{v}}_{KF}-\left(\omega_{rl}\cdot r_{dyn}+\omega_{rr}\cdot r_{dyn}\right)\cdot 0.5$$
(43)$$r_{16}={\hat{a}}_{KF}-a_{x}$$

In addition, from (Equation 22) and (Equation 13), we could also obtain another two residuals for the side-slip angle:(44)$$r_{17}={\hat{\beta }}_{KF}-{\hat{\beta }}_{\delta }$$
(45)$$r_{18}={\hat{\beta }}_{KF}-{\hat{\beta }}_{\dot{\psi }}$$

### 3.6. Model of Tie-Rod Force

In this work, the forces on the left and right tie rods are estimated together as the total tie-rod force on the rack. In a normal driving situation, there is a linear relationship among the steering-wheel torque $M_{S}$, the lateral forces of the left and right front wheels $F_{yfl}$, $F_{yfr}$, the concentrated friction torque $M_{FR,S}$ and the total tie-rod force. However, in practice, these two forces $F_{yfl}$ and $F_{yfr}$ are unknown and difficult to obtain under complex test conditions. Therefore, we replace the sum of these two forces with the product of the vehicle mass and the lateral acceleration. Therefore, a model can be created to approximate the total force on the tie rod:(46)$$F_{TieRod_{sum}}=a_{4}\cdot m\cdot a_{y}+a_{5}\cdot \left(M_{S}+M_{FR,S}\right)$$
where $a_{4},a_{5}$ are the coefficients determined in the simulation. Then, a new residual is:(47)$$r_{19}={\hat{F}}_{TieRod_{sumay}}-F_{TieRod_{sum}}$$

### 3.7. Model Validation

Since the main content of this method is to compare the signal predicted by the model with the signal measured by the sensor to monitor the condition of the sensor, the accuracy of the model is essential. Therefore, we must first validate the model. The simulated data that is collected from the CarMaker would be here in place of the actual sensor signal. The CarMaker platform is a fully fledged virtual driving environment developed by IPG AUTOMOTIVE, and it can offer a wide range of applications, from offline operation to hardware-in-the-loop (HIL) tests. The CarMaker vehicle model is extremely realistic, since it comprises models of the driving system, the tires, the steering system, the suspension, the sprung body, the driver, the road, and the environment [22]. Therefore, the car model built in CarMaker can actually be regarded as a virtual car. All simulation results in this article use the same car model, which was created in CarMaker based on the validated parameters of the real car in project “SmartLoad” [12]. The relevant vehicle parameters are listed in Table A1 in Appendix A.

Figure 5 shows the comparison of simulated and measured values for the steering-wheel angle, the steering-wheel torque, the yaw rate, the left rear wheel velocity, the right rear wheel velocity and the tie-rod force. The blue line shows the measured values, the other colours show the simulated values, i.e., the sensor signals predicted by the model. It can be seen that the results are extremely precise. A relatively accurate model will also facilitate subsequent anomaly detection, because in the event of a fault, the difference between the predicted value and the measured value is obvious.

### 3.8. Symptom Generation

The residual value generated in the fault detection can also be called the symptom of the fault. We must first perform a series of processes on it. Due to a series of influences, such as model accuracy or extreme driving operations, the residual value will generally have a fluctuation range, and the variable residual range under different parameters is also different. Therefore, the residual is generally first analyzed, and a threshold within a normal range will be given. Depending on the threshold, anomaly detection usually also exhibits different degrees of sensitivity. When a residual threshold is given, there are usually two types of thresholds—static thresholds and dynamic thresholds. The establishment of the static threshold is relatively simple, but it is not suitable for changing situations, while the establishment of the dynamic threshold is relatively complex, but it can be more suitable for complex situations. In this article, we first use the static threshold because of the simplicity of its implementation, and since the system we designed in this article is only for the detection of anomalies, the static threshold can already be well implemented. A simple example is shown in Figure 6. The red dotted line represents the normal threshold of the residual of the current steering angle, and the range beyond the red dotted line is regarded as an abnormal situation.

After a series of tests, we determined the normal threshold range for each residual value, set the residual within its corresponding threshold range to 0, and set the residual exceeding its corresponding threshold to 1. From this, we obtain a symptom table for further state analysis of the six sensors: the steering-wheel angle sensor $\delta_{S}$, the steering-wheel torque sensor $M_{S}$, the yaw rate sensor $\dot{\psi }$, the velocity of left rear wheel $v_{rl}$, the velocity of right rear wheel $v_{rr}$ and the tie-rod force sensor $F_{TieRod}$.

For example, when an exemplary sensor fails, it can be clearly seen from Figure 7 that between 20 and 40 s, each residual value presents different warning states: 0 means there is no abnormal situation, and 1 means there is an abnormal situation. Overall, the alarm starts at 20 s and ends at 40 s. From this, it can be judged that the sensor system has an abnormal situation between 20 and 40 s. When the monitoring system needs to be further diagnosed to determine which sensor is faulty, a series of judgment rules need to be made according to Table 1. As can be seen from the table, the faulty symptoms of the two speed sensors are the same, but if the positive and negative residuals are analyzed using the judgment criteria listed below, it can also be concluded which speed sensor is faulty.

*IF $r_{1}$ is normal AND $r_{2}$ is abnormal AND $r_{3}$ is abnormal, THEN steering-wheel angle sensor is faulty*.*IF $r_{1}$ is normal AND $r_{2}$ is normal AND $r_{3}$ is abnormal AND $r_{4}$ is normal AND $r_{17}$ is normal AND $r_{18}$ is normal AND $r_{19}$ is abnormal, THEN steering-wheel torque sensor is faulty*.*IF $r_{1}$ is normal AND $r_{2}$ is abnormal AND $r_{3}$ is normal AND $r_{4}$ is abnormal AND left rear wheel-speed sensor is normal AND right rear wheel-speed sensor is normal, THEN yaw rate sensor is faulty*.*IF $r_{7}$ is abnormal AND $r_{9}$ is abnormal AND $r_{11}$ is abnormal AND $r_{13}$ is abnormal AND $r_{10}$ is abnormal AND $r_{14}$ is abnormal AND $r_{14}$ is negative, THEN right rear wheel-speed sensor is faulty*.*IF $r_{7}$ is abnormal AND $r_{9}$ is abnormal AND $r_{11}$ is abnormal AND $r_{13}$ is abnormal AND $r_{14}$ is abnormal AND $r_{10}$ is abnormal AND $r_{10}$ is negative, THEN left rear wheel-speed sensor is faulty*.*IF $r_{1}$ is normal AND $r_{2}$ is normal AND $r_{3}$ is normal AND $r_{4}$ is normal AND $r_{17}$ is normal AND $r_{18}$ is normal AND $r_{19}$ is abnormal, THEN tie-rod force sensor is faulty*.

According to the judgment criteria given above, we can obtain the result as shown in Figure 8, i.e., between 20 and 40 s, the left rear wheel-speed sensor fails (purple line). At the same time, the status of the other five sensors is normal, and the results correspond to the other lines, which coincide with the blue line. From the above fault–symptom table, it can be inferred that there are more than six rules. After the verification of the diagnosis rate, these six rules have the highest diagnostic rate and efficiency, so they are written here.

### 3.9. Sensor Reconfiguration

The results of the condition monitoring of the sensors help correct the sensor data input to the actuator condition-monitoring process as Figure 9 shows. When the sensor is running normally, we use the original sensor data in the actuator condition monitoring, but when the sensor condition monitoring shows abnormalities, we use the sensor value estimated by the model in the next actuator monitoring.

For example, the demo vehicle is driving on the test route. A total-loss fault occurs on the left rear wheel-speed sensor at 20 s and lasts for 20 s. The sensor value with no fault is shown as the blue lines in Figure 10, the actual sensor-measured value with fault is shown as the red line, the model estimated value is shown as the yellow line, and then as input to the followed system. The corrected speed sensor value will be very close to the original no-fault value despite the fault. This process can ensure that the system will always have relatively accurate sensor values for subsequent control and actuator status detection, and minimize the impact of sensor failures on the entire system.

## 4. Model-Based Actuator Condition Monitoring

Before introducing the method of condition monitoring of the actuator system, the actuator structure of the demonstrator is briefly introduced. The structure of the actuator system in the demonstrator vehicle is shown in Figure 11. It is a 1:1.5 scale vehicle with Ackermann steering and wheel-selective electric drives on the steered axle. The rear axle of the vehicle is designed as a non-driven axle. The two wheels of the steered front axle are driven individually ($EM_{A1}$ and $EM_{A2}$). The steering system consists of a steering column, a steering gear (Z) with rack and pinion and the tie rods. Two steering actuators, which are placed on the steering column, are available for controlling the steering angle. One of the steering motors serves as a redundancy. The wheel-selective drive enables independent generation of a drive torque on each wheel of the front axle. By distributing the drive torques to the two wheels, a steering torque can be generated, which can be used to control a steering angle as the second redundancy of the steering function.

The method for condition monitoring of the actuator is to calculate the ideal driving state of the car through the control signals, and compare it with the actual driving state of the demonstrator calculated by the sensor signal to obtain the current state of the actuator. In addition to the sensor signals introduced in the previous section, what we can use for condition monitoring of the actuators includes the target speed, the target steering-wheel angle, and the target torque of the actuator given by the controller.

The model used in the condition monitoring of the actuator is actually the same as the model used in the sensor condition monitoring. The only difference is that the input to the model is the current measurement value of the sensor during the fault detection of the sensor, which is the so-called car’s actual state. However, in the condition monitoring of the actuator, the input to the model is the target state of the vehicle, i.e., the theoretical state when the actuator does not fail, and the current state of the vehicle actuator can be detected through the difference.

First, we are still going to use the linear single-track model from Section 3.3 for the estimation of the side-slip angle and yaw rate. This has been introduced in the previous paragraph and will not be repeated here. To calculate the desired driving state of the car in the current ideal state, we need to use the target speed and target angle as the model inputs for the calculation of the ideal side-slip angle and yaw rate. From this model, we could obtain two values related to the condition monitoring of the steering motor:(48)$$\begin{array}{cc} \; & r_{1a}=\Delta_{\dot{\psi }}={\dot{\psi }}_{ideal}-{\dot{\psi }}_{actual}\\ \; & r_{2a}=\Delta_{\beta }=\beta_{ideal}-\beta_{actual} \end{array}$$

In addition to the model mentioned in Section 3.3, we will also use the models mentioned in Section 3.4. The difference here is used to calculate the ideal longitudinal and lateral acceleration with the target velocity and ideal yaw rate, as well as the ideal side-slip angle that was calculated before. From this model, we could also obtain two values related to the condition monitoring of the steering motor and driving motor:(49)$$\begin{array}{cc} \; & r_{3a}=\Delta_{a_{x}}=a_{x_{ideal}}-a_{x_{actual}}\\ \; & r_{4a}=\Delta_{a_{y}}=a_{y_{ideal}}-a_{y_{actual}} \end{array}$$

The differences between target values and actual measured values of the steering-wheel angle and steering-wheel torque can be used to detect the condition of the actuators:(50)$$\begin{array}{cc} \; & r_{5a}=\Delta_{\delta_{S}}={\delta_{S}}_{target}-{\delta_{S}}_{actual}\\ \; & r_{6a}=\Delta_{M_{S}}=M_{S_{target}}-M_{S_{actual}} \end{array}$$

The actual measured value of velocity *v* can be generated from two rear wheel-speed sensors. According to the condition monitoring of the sensor, it is decided whether to use the measured value or the corresponding predicted value.
(51)$$v_{actual}=\left(\omega_{rr}+\omega_{rl}\right)\cdot \frac{r_{0}}{2}$$

Moreover, we could also use the same model from Section 3.6 to estimate the ideal force on tie rod $F_{TieRod_{ideal}}$. The only difference here is to use the ideal lateral acceleration and target steering torque.

Then, we could also obtain the differences between target values and actual measured values as:(52)$$\begin{array}{cc} \; & r_{7a}=\Delta_{v}=v_{target}-v_{actual}\\ \; & r_{8a}=\Delta_{F_{TieRod}}=F_{TieRod_{ideal}}-F_{TieRod_{actual}} \end{array}$$

When each actuator fails, the symptoms mentioned above will also change accordingly. From this, we can also diagnose which actuator is faulty through a symptom rule table. Here, for a clearer explanation, we name casea when the steering actuator fails and caseb when the driving actuator fails.

### 4.1. Case a: Failure of Steering Motor

As shown in Figure 12, when both steering motors fail, the corresponding fault symptoms change. It can be clearly seen from the figure that after 104 s, the corresponding eight symptoms have an obvious change compared to the previous time when no failure occurred. We have introduced the static threshold in Figure 6; we also use the same method here, setting the part exceeding the threshold as abnormal as 1, and the part not exceeding the threshold as normal as 0. Then we could obtain a symptom regularity Table 2 that is similar to the sensor condition monitoring.

It can be seen from Table 2 that the symptom rules are the same for the two steering motors, and because the sensor on the steering rod measures the sum torque of the two steering motors, it can only be determined from the dynamic state quantity whether the steering motor is faulty. Specifically, to distinguish which steering motor, then it is necessary to use the respective target torques of the two steering motors, and in the event of a fault, the target torques will also change accordingly. A simple schematic diagram of the symptoms changing is shown in Figure 13. It can be seen that at the position of 104s, eight fault symptoms have alarms of different degrees. The abnormality of the eight symptoms corresponds to the fault of the steering motor. Through further analysis of its target torque, it can be obtained as shown in Figure 14 that the two steering motors have failed in the current situation.

### 4.2. Case b: Failure of Driving Motor

The method used to detect the failure of the drive actuator is actually similar to that of the steering actuator. It can be seen from Figure 15 that when the drive actuator fails, its eight symptoms show different trends, and the force on the tie rod and the steering-wheel torque have obvious changes, while the other symptoms are not as obvious as the situation with the steering motor. We also set different static thresholds for different symptoms, and set the ones that exceed the thresholds to 1, which is abnormal, and those that are within the thresholds to 0, which is normal. From the difference in representation under different conditions, we can easily detect the current state of the actuator through a symptom regularity Table 3 that is similar to the sensor condition monitoring.

As for the left or right side of the drive motor failing, the symptom from the tie-rod force will change significantly. To distinguish the fault that occurs on the left side or the right side, we can use the positive and negative values of the symptom to judge. It is also a simple schematic diagram of the symptoms changing, as shown in Figure 16. It can be seen that when the driving actuator is abnormal, the symptoms alarming will be significantly less than those associated with the abnormal situation of the steering motor. From this, it can be judged that the motor is abnormal after 104 s. Through further analysis of the tie-rod force symbol, it can be concluded that the left motor is abnormal, as shown in Figure 17.

## 5. Performance Evaluation

Now we want to evaluate the performance of the method discussed in this article. The two paths we mainly use here are derived from project SmartLoad [12]. Compared with the general test path, these two paths are more complex and cover a lot of extreme driving situations, which is good for testing the robustness of this method. The two paths are shown in Figure 18 and Figure 19. In the test, the data we use is obtained from the simulation in CarMaker with the car model of the project, which is used to replace the data of the sensors on the real car. In this paper, the tested sensor fault types are divided into four types, namely: total-loss fault, positive offset, negative offset and outlier. Total-loss fault means that the sensor has no function at all, and the displayed data are 0; positive offset fault means that the data displayed by the sensor has a positive deviation from the actual data; negative offset fault means that the sensor data have a negative deviation from the actual data, and outlier fault refers to a large outlier in the sensor data over a very short time such as one second. The monitoring result is to be able to detect abnormalities as the acceptance standard. That is to say, the method introduced in this article only detects the abnormalities of the sensors, and the detection result for the abnormality is 1, without specific fault classification, while the result is 0 when the sensor is normal.

For sensors, we have completed 48 tests. They are for six sensors, and each sensor has four fault types. The tests were performed under the two paths shown in Figure 18 and Figure 19, and each test lasts 60 s. The method we use here to evaluate performance is performance metrics for classification problems.

The simplest performance metric for classification is the accuracy rate. This can be done with the formula: (53)$$ACC=\frac{correct\;predictions}{all\;predictions}$$

The accuracy rates for the sensor monitoring are shown in Figure 20. From the accuracy rate in the figure, we can see that the accuracy rate of the abnormal condition monitoring of the sensors is outstanding and can reach more than 0.95.

Although the overall accuracy rate of the tests is already very high, if it is aimed at a single test, there are still problems. For example, in the case of a total-loss fault of the IMU sensor, the accuracy rate is only 0.86. Figure 21 shows the diagnostic results (blue line) and the actual state of the fault (red line). It can be seen from the figure that no fault is detected between 35 and 40 s, and the same problem still exists between 24.5 and 25.5 s. To find the reason for this, we can look at the original sensor data. As shown in Figure 22, the blue line refers to IMU sensor data with no fault, and the red line refers to IMU sensor data with total-loss fault between 20 and 40 s. From Figure 22, we can easily see that the time range where the fault detection fails to refer to the time range where the deviation between the normal data (blue line) and the faulty data (red line) is very small. For example, at about 25 and 35 s, the actual yaw rate is already around 0, and the sensor reading of the total-loss fault is 0, so it is difficult to diagnose the fault in this state. Moreover, since the static threshold is used in this paper, it cannot change with different driving conditions, which provides an obstacle to the detection. It can be seen from Figure 23 that the residual values used to judge the IMU fault at 25 s and at 35 s had no warnings appear.

For the condition monitoring of the actuator, we conducted nine tests, and under three paths. Another one is shown in Figure 24, also from the project SmartLoad [12]. Each test lasts 200 s. To evaluate the performance, the fault signal is imported into the simulation. In these nine tests, only the completely damaged situation is in consideration, and the output torque is completely 0.

For performance evaluation, we still use the above accuracy formula. It can also be seen from Figure 25 that the method studied in this article also has a high accuracy for condition monitoring of the actuator.

However, there are also problems in the fault diagnosis of the actuator. For example, in Figure 26, the diagnosis result shows the fault 1.8 s slower than the actual fault occurs. We can also see in Figure 27 the residual that was used for diagnosis: the position where the alarm occurs is consistent with the position where the diagnosis result shows the fault, i.e., it is later than the actual situation. To find the reason, we need to pay attention to the actual torque. As can be seen from Figure 28, the actual torque of the motor is already very small in the three seconds before the fault, which leads to the actual state of the car and the ideal state status being not much different, and leads to subsequent misdiagnosis.

It can be seen from the above that there is a problem in the diagnosis of both sensors and actuators, i.e., if the state of the car itself is relatively low speed and has small steering angle when a fault occurs, then the method in this paper is likely to judge this state as a no-fault state.

All the results shown in this article are simulated results, i.e., the car model built in CarMaker based on the validated parameters of the real car is regarded as a virtual car, and its driving states in the virtual environment are regarded as the sensor measurement values. On the one hand, this is due to the lack of current real vehicle data; on the other hand, due to the limitations of experiments, it is impossible to inject faults into the motors of real vehicles. Therefore, the motor faults of real vehicles are actually adjusted to 0 artificially, which is different from that in the simulation. In order not to confuse readers further, there is no display of real vehicle data here. However, it will be written in the author’s next article, which is aimed at the comparison and evaluation of different diagnostic methods, which will involve the verification of real vehicle data.

## 6. Conclusions

In this work, a model-based method has been developed for the holistic fault detection and diagnosis system to monitor the condition of sensors and actuators. The monitored sensors in this article are the left rear wheel-speed sensor, right rear wheel-speed sensor, steering-wheel angle sensor, steering-wheel torque sensor, yaw rate sensor, tie-rod force sensor, and the monitored actuators are two steering motors, and two driving motors that were equipped in the demonstration vehicle. The driving data will be produced by CarMaker, and the monitoring system model is built in Simulink. It can realize real-time monitoring by connecting with the car model in CarMaker (software in the loop), or obtain simulation data from CarMaker first, and then perform offline state detection in Simulink.

The basic idea of the fault detection method in this article is to generate the residuals as the symptoms through the mathematical models of the vehicle. After the sensor residuals are obtained, through the analysis of the changes in symptoms in different abnormal situations, the detection rules are summarized, and the status of each sensor is determined through a simple logical relationship. In addition, the results are used to correct the sensor data and ensure that the sensor values input to the subsequent system are fault-free. The difference between the target value and the corrected sensor value is used as the monitoring of the actuator, and it also determines the state of each actuator through a series of simple logic relationships. The binary logic rules used to detect states in this article are very simple and less time-consuming to use than other diagnostic methods, and this method is very effective for detecting abnormal and normal conditions of current sensors and actuators.

However, for the analysis of specific fault situations, such as what kind of fault occurred in the specific sensor, and how much torque loss the actuator produced, it cannot be effectively dealt with. In future research, the author will continue the current problem to evaluate and compare different fault-diagnosis methods. A systematic comparison and discussion of the diagnostic capabilities of different fault-diagnosis methods are carried out. In subsequent articles, complex scenarios, such as the simultaneous fault of multiple sensors, will also be investigated. These different fault conditions can first generate symptoms through the fault–symptom generation method described in this paper, and then use different fault-diagnosis methods to further analyze the complexity and concrete situation of the fault. In addition to this, the method in this article will continue to be used and validated with other project vehicles. The validation of the method in this paper is now limited to the simulation data. Due to the complexity of injecting faults into the motor in the real vehicle, there are few real vehicle experimental data, and the method of fault injection is also different to that in the simulation. Therefore, in this paper, it is just validated using the simulated data. Further research on this issue will also be carried out in the author’s subsequent articles.

## Figures and Tables

**Figure 1 sensors-23-00887-f001:**
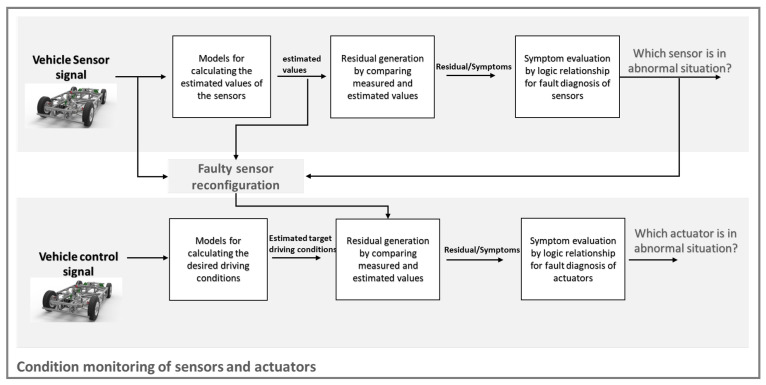
Scheme of condition monitoring of sensors and actuators.

**Figure 2 sensors-23-00887-f002:**
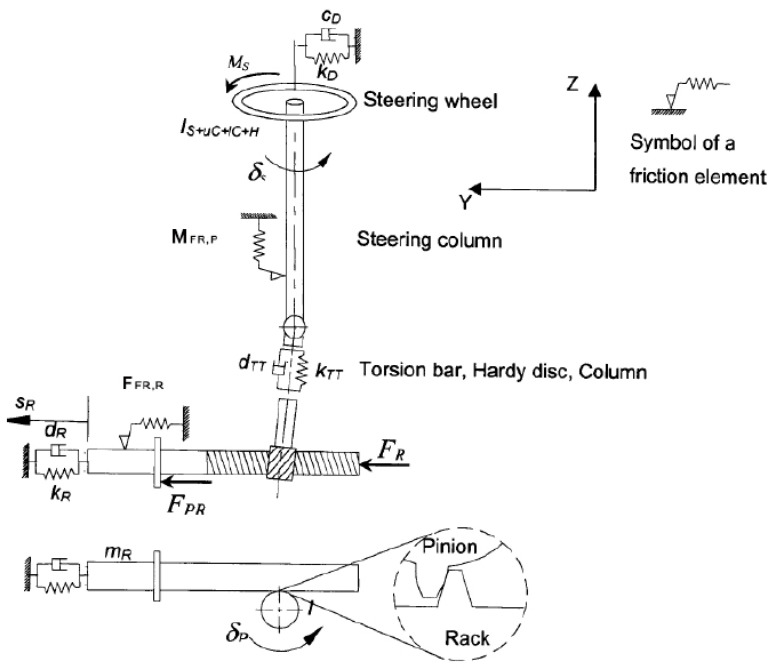
2-DOF steering model. Reprinted/adapted with permission from Ref. [20]. 2006, Peter Pfeffer.

**Figure 3 sensors-23-00887-f003:**
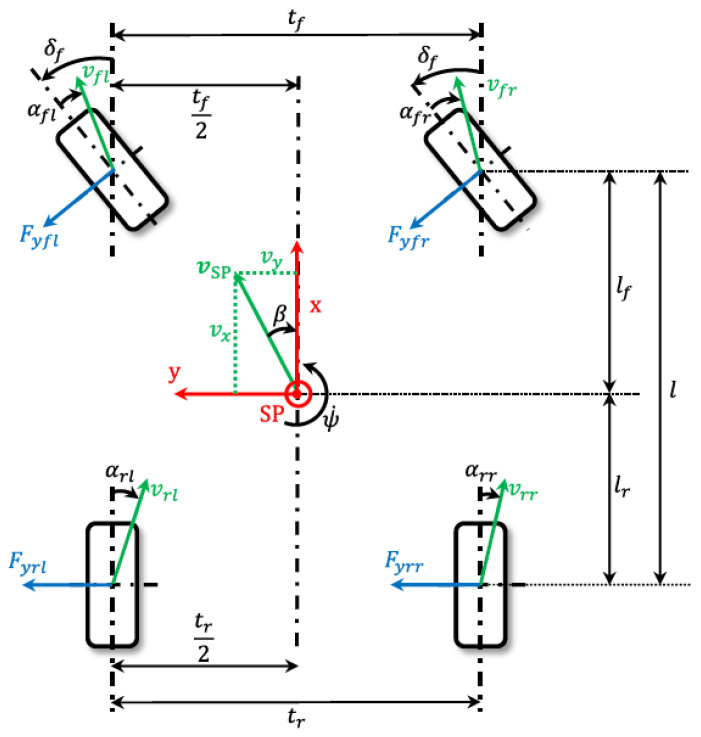
Two-track model [21].

**Figure 4 sensors-23-00887-f004:**
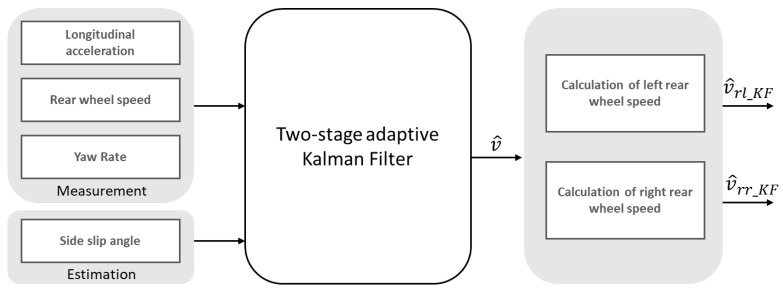
Structure of the estimation of rear wheel speed.

**Figure 5 sensors-23-00887-f005:**
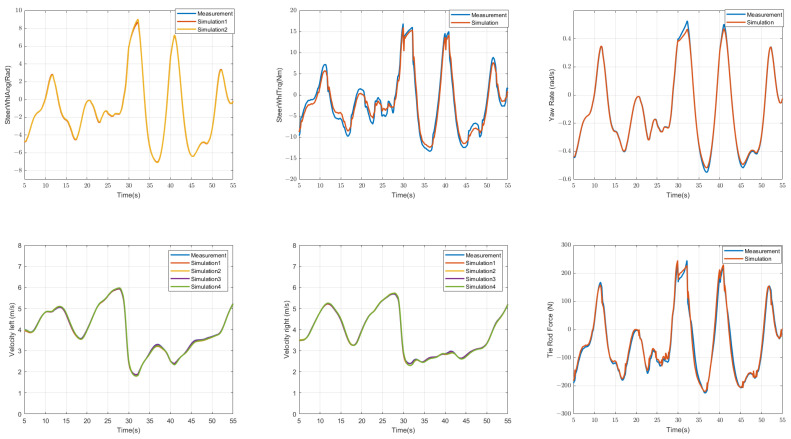
Results of model validation: Comparison of the measured data (blue line) and the simulated data (lines with other colours) of the steering-wheel angle, the steering-wheel torque, the yaw-rate, the left rear wheel velocity, the right rear wheel velocity and the tie-rod force.

**Figure 6 sensors-23-00887-f006:**
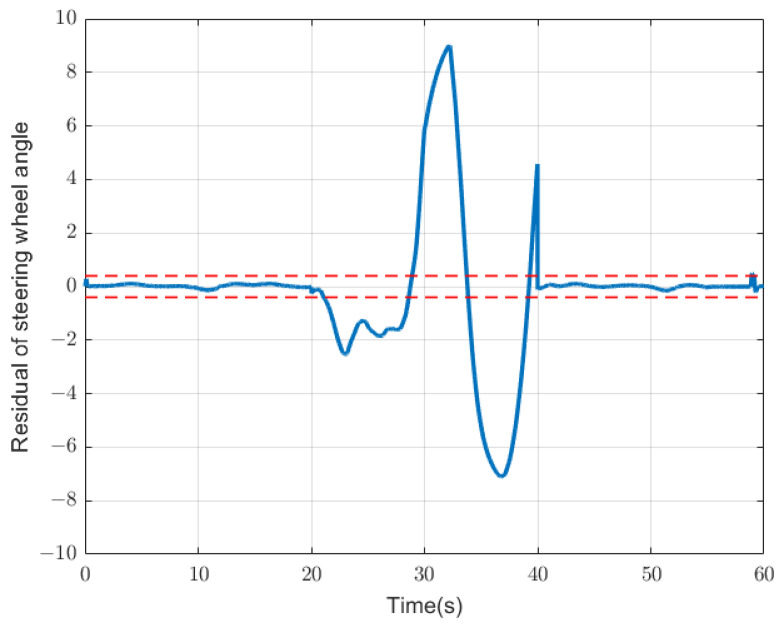
An example of the static threshold: the blue line represents the residual value of the steering-wheel angle, and the red dotted line represents the static threshold. Between 20 and 40 s, the blue residual value obviously exceeds the static threshold and is considered abnormal.

**Figure 7 sensors-23-00887-f007:**
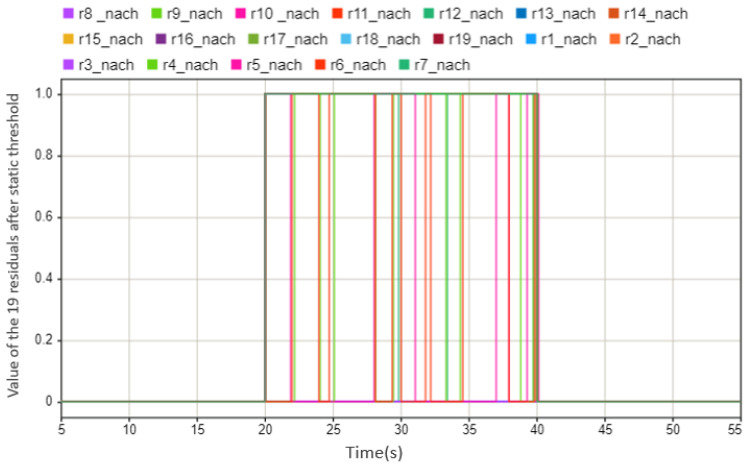
The values of the 19 residuals after threshold. This case shows that the state of sensors is abnormal between the 20 s and 40 s (0 means there is no abnormal situation, 1 means there is an abnormal situation).

**Figure 8 sensors-23-00887-f008:**
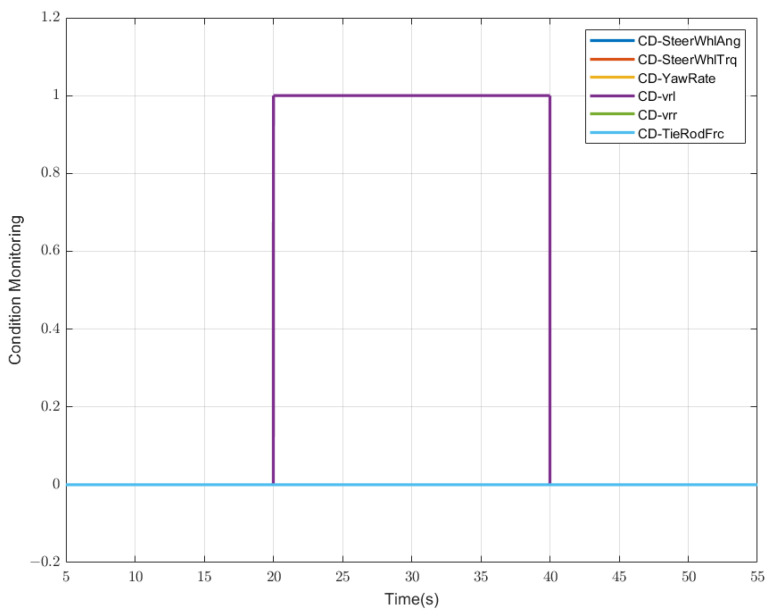
Result of sensor condition monitoring: the case is that the left rear wheel velocity fails between the 20 s and 40 s (“CD” is an abbreviation for “condition”, “0” refers to the normal state, “1” refers to the faulty state).

**Figure 9 sensors-23-00887-f009:**
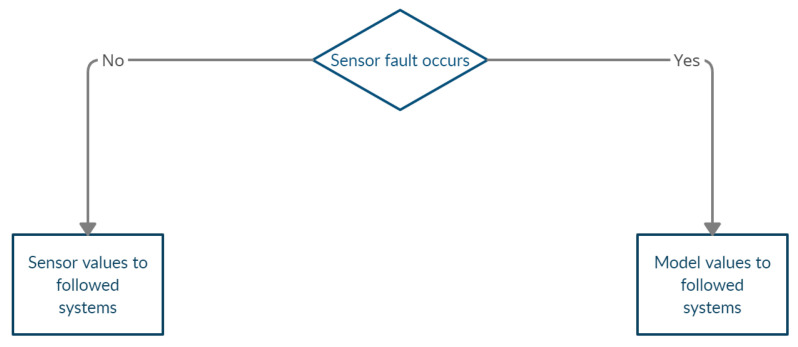
Process of the sensor reconfiguration.

**Figure 10 sensors-23-00887-f010:**
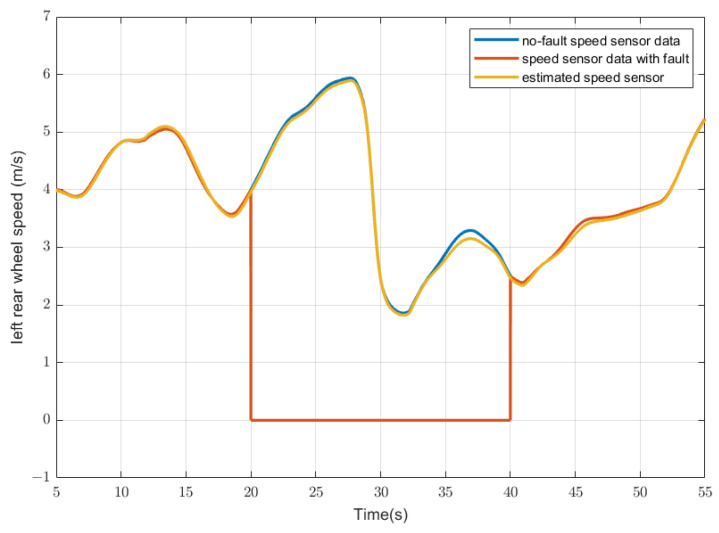
An example of sensor reconfiguration. The case is that the left rear speed sensor has a total-loss fault between 20 and 40 s. (Blue line: displayed sensor data with no-fault; red line: actual displayed sensor data with a fault; yellow line: estimated sensor data, which replaced the actual displayed sensor data).

**Figure 11 sensors-23-00887-f011:**
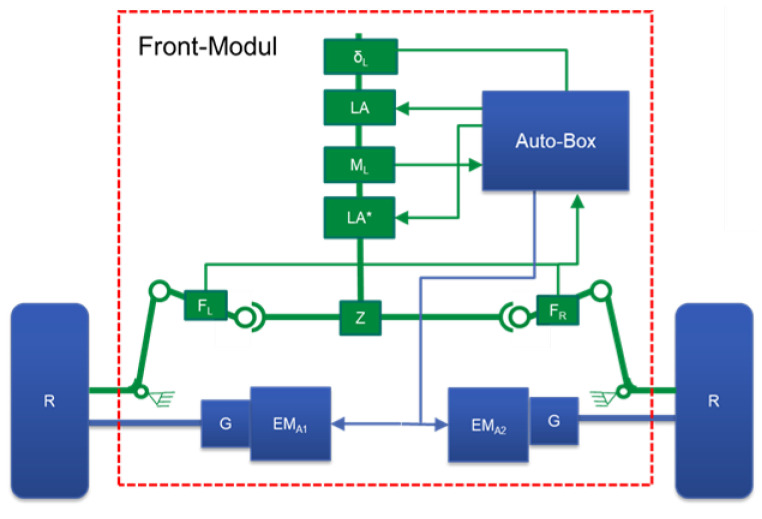
Front model structure of the demonstrator [12].

**Figure 12 sensors-23-00887-f012:**
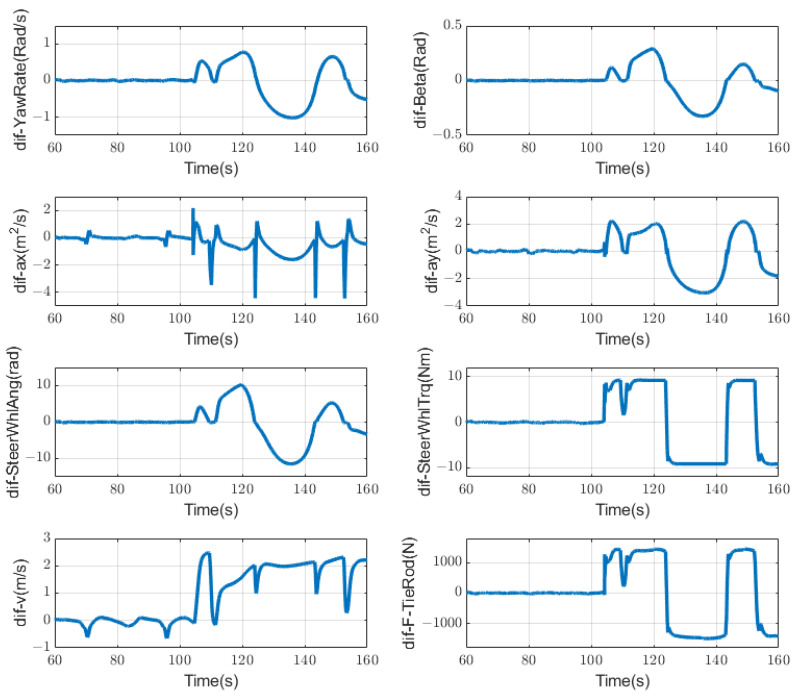
Symptoms changing trend when steering motor fails at 104 s.

**Figure 13 sensors-23-00887-f013:**
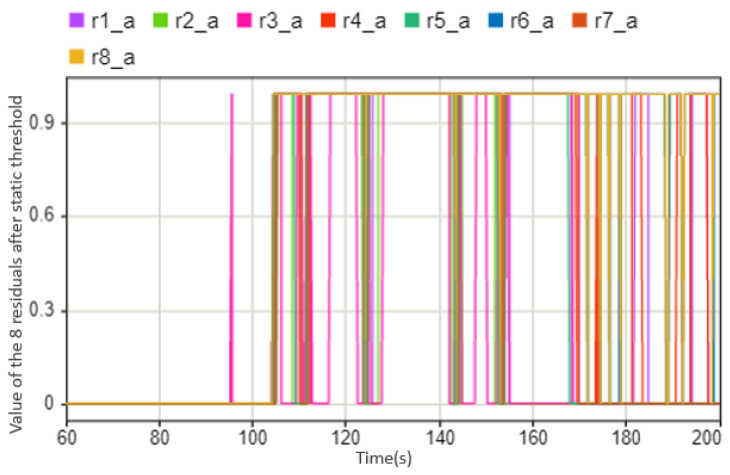
The values of the eight residuals after threshold, this case shows that when steering motor fails between 104 s and 200 s (0 means there is no abnormal situation, 1 means there is an abnormal situation).

**Figure 14 sensors-23-00887-f014:**
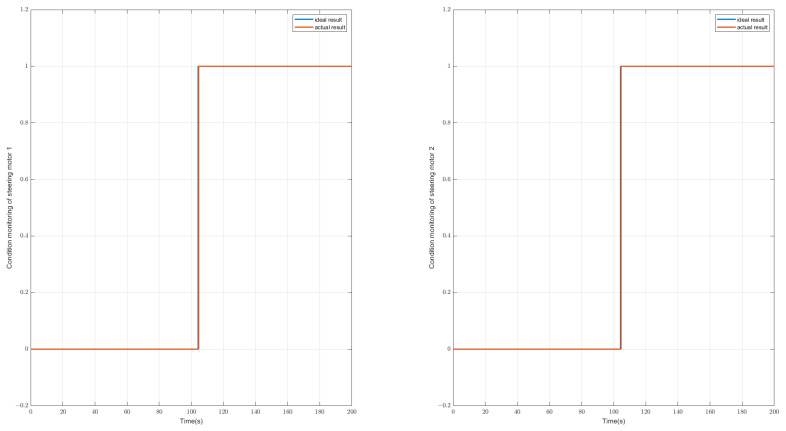
Diagnostic result of condition monitoring of the steering motors, this case shows that the steering motor 1 and steering motor 2 fail at 104 s. (Left: results for motor 1; Right: result for motor 2; Blue line: real state of the steering motor; Red line: diagnostic state of the steering motor).

**Figure 15 sensors-23-00887-f015:**
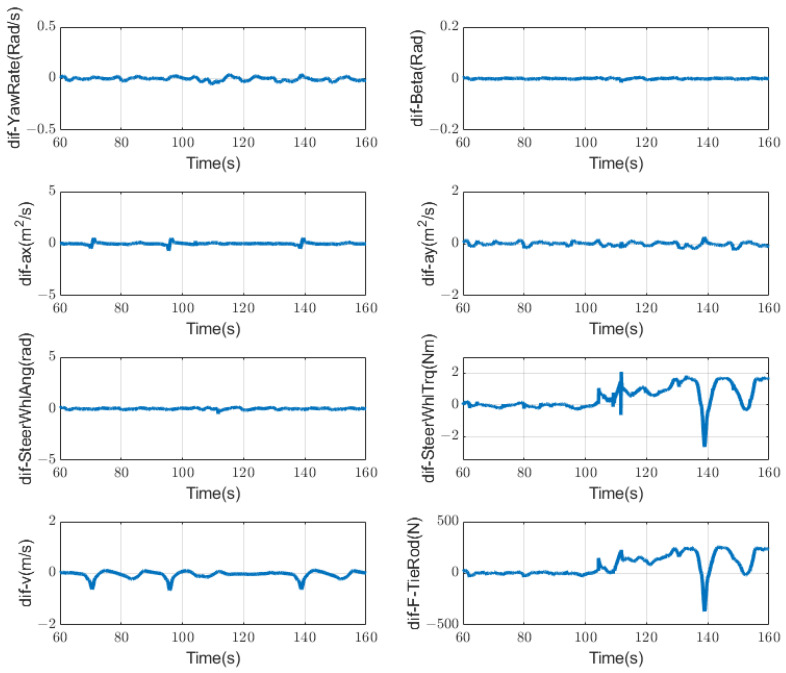
Symptoms changing trend when driving motor fails at 104 s.

**Figure 16 sensors-23-00887-f016:**
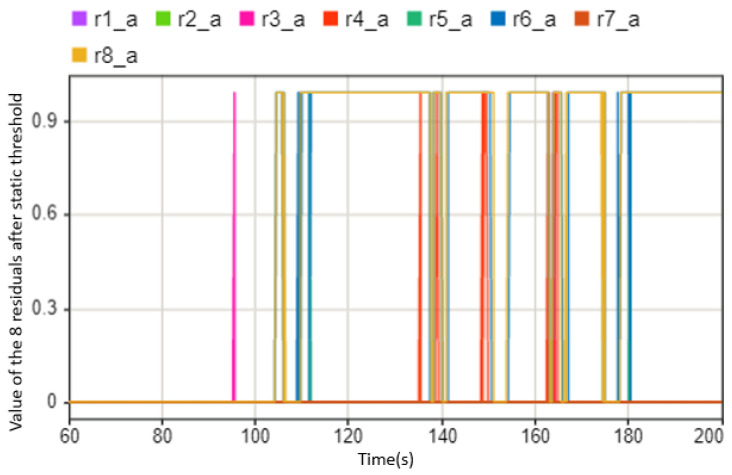
The values of the eight residuals after threshold. This case shows when driving motor fails between 104 s and 200 s (0 means there is no abnormal situation, 1 means there is an abnormal situation).

**Figure 17 sensors-23-00887-f017:**
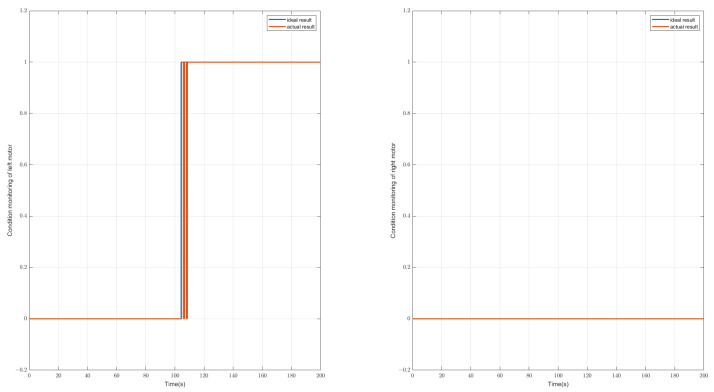
The diagnostic result of condition monitoring of the driving motors. This case shows that the left driving motor fails at 104 s. (left: results for left motor, right: result for right motor, blue line: real state of the driving motor, red line: diagnostic state of the driving motor).

**Figure 18 sensors-23-00887-f018:**
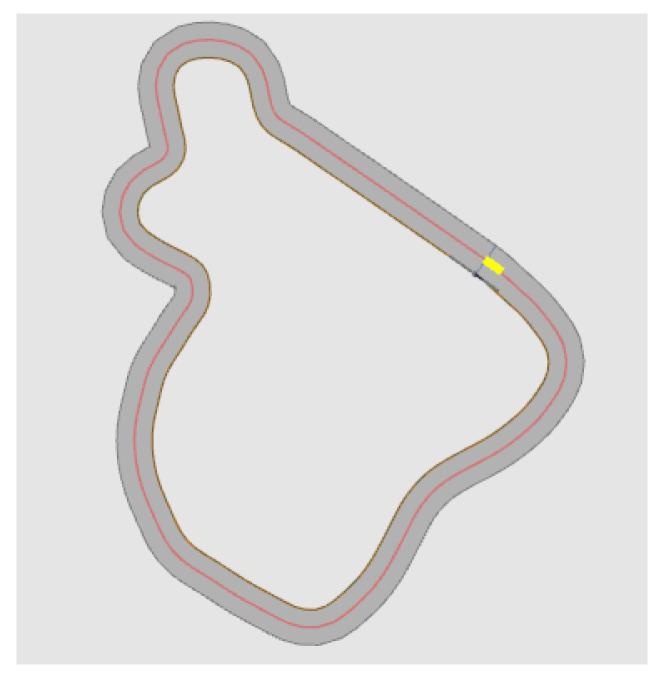
Short round driving test circuit, the red line represents the route, and the yellow square represents where the car starts driving.

**Figure 19 sensors-23-00887-f019:**
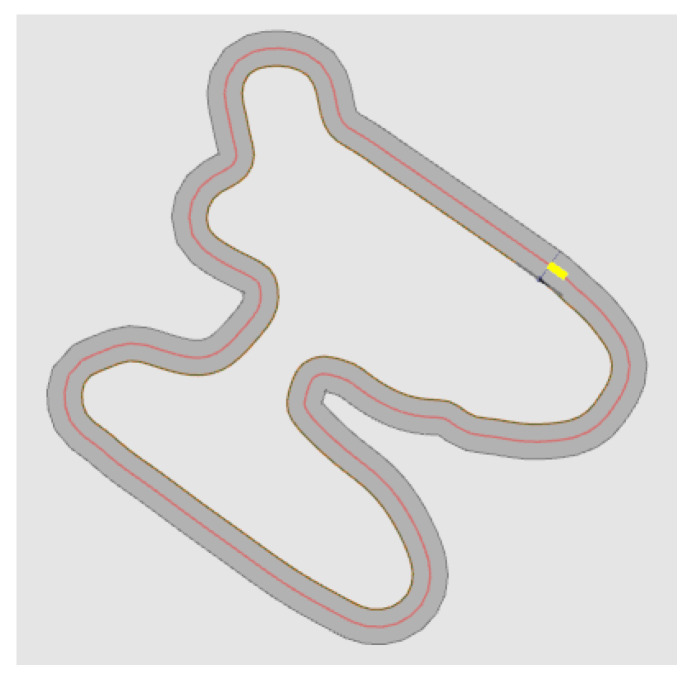
Long round driving test circuit, the red line represents the route, and the yellow square represents where the car starts driving.

**Figure 20 sensors-23-00887-f020:**
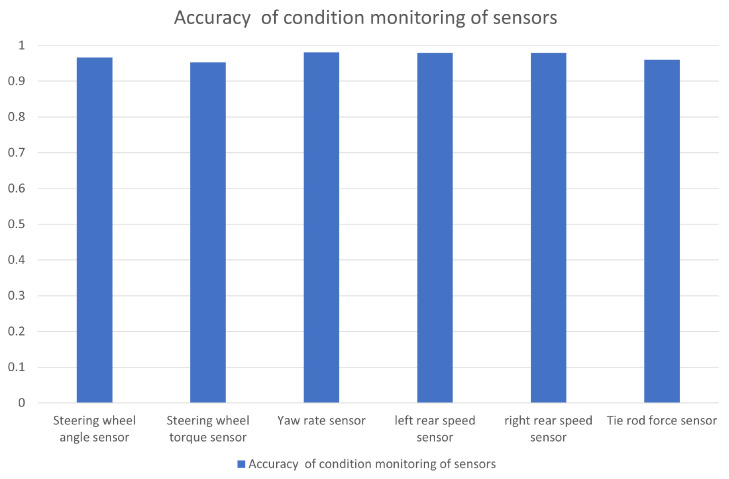
Accuracy of condition monitoring of different sensors.

**Figure 21 sensors-23-00887-f021:**
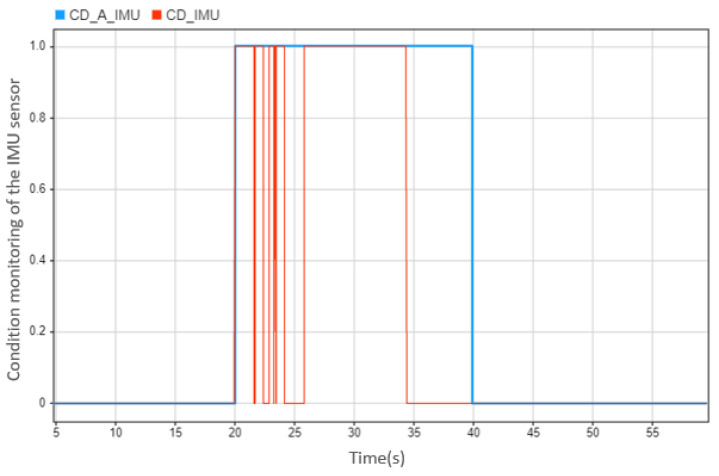
The diagnostic result of condition monitoring of IMU sensor: the case is that the IMU sensor fails between 20 s and 40 s, but the diagnostic result shows the failure is only between 20 s and 34.4 s (“CD” is an abbreviation for “condition”, “0” refers the normal state, “1” refers the faulty state, the blue line refers to the real state of the IMU sensor, the red line refers to the diagnostic state of the IMU sensor).

**Figure 22 sensors-23-00887-f022:**
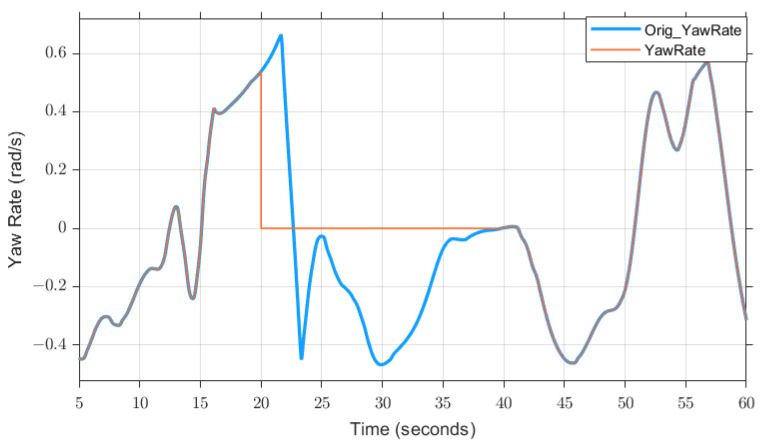
IMU sensor data, which fails between 20 s and 40 s (the blue line refers to the displayed data when the sensor does not fail, the red line refers to the displayed data when the sensor fails).

**Figure 23 sensors-23-00887-f023:**
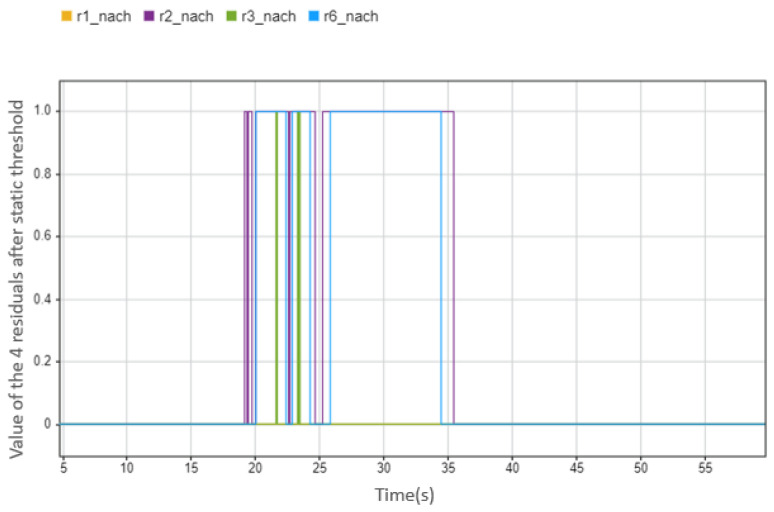
The values of the four residuals after threshold, this case is that when IMU sensor fails between 20 s and 40 s (0 means there is no abnormal situation, 1 means there is an abnormal situation).

**Figure 24 sensors-23-00887-f024:**
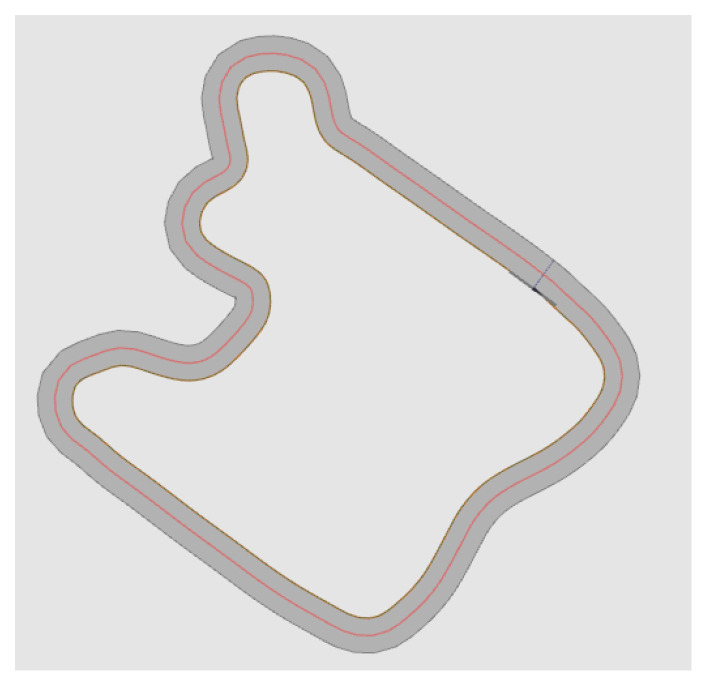
Third round driving test circuit.

**Figure 25 sensors-23-00887-f025:**
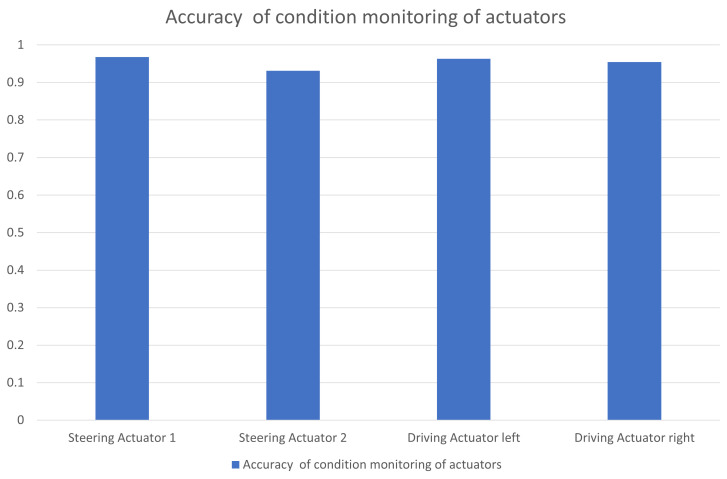
Accuracy of condition monitoring of different actuators.

**Figure 26 sensors-23-00887-f026:**
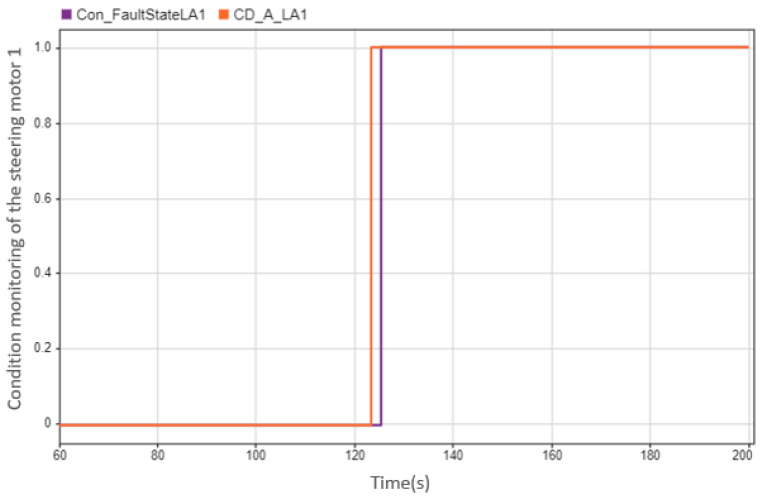
Result of condition monitoring of steering motor 1. Diagnostic result of condition monitoring of the steering motor 1: the case is that the steering motor 1 fails at 123.5 s, but the diagnostic result shows the failure is at 125.3 s (“CD” is an abbreviation for “condition”; “0” refers the normal state; “1” refers the faulty state; the orange line refers to the real state of the steering motor 1; the purple line refers to the diagnostic state of the steering motor 1).

**Figure 27 sensors-23-00887-f027:**
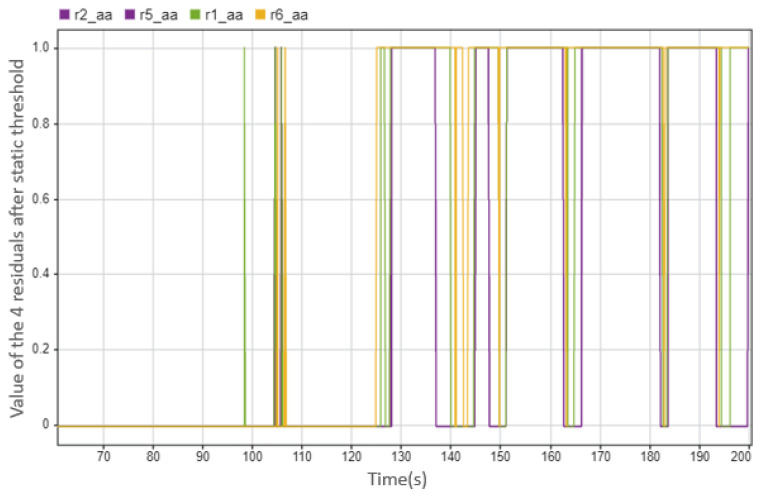
The values of the four residuals after threshold. This case shows when the steering motor 1 fails at 123.5 s (0 means there is no abnormal situation, 1 means there is an abnormal situation).

**Figure 28 sensors-23-00887-f028:**
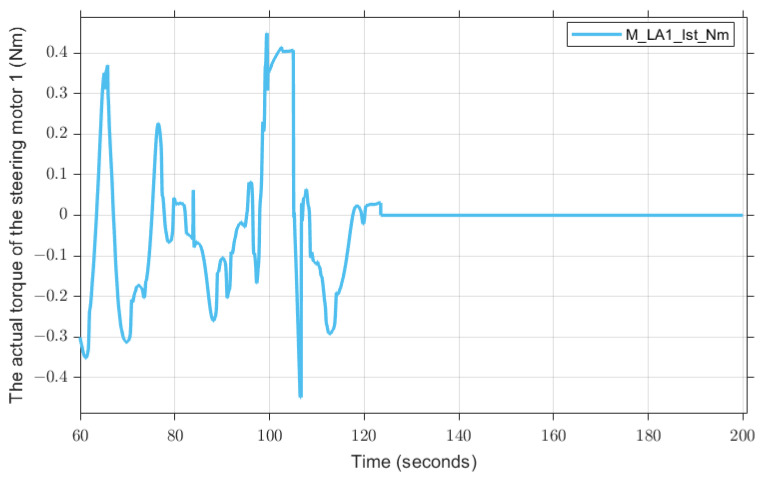
Actual steering torque of the steering motor 1, which fails at 123.5 s.

**Table 1 sensors-23-00887-t001:** Fault–symptom relationships of $r_{1}$ to $r_{8}$ for condition monitoring of six sensors: the steering-wheel angle sensor $\delta_{S}$, the steering-wheel torque sensor $M_{S}$, the yaw rate sensor $\dot{\psi }$, the velocity of left rear wheel $v_{rl}$, the velocity of right rear wheel $v_{rr}$ and the tie-rod force sensor $F_{TieRod}$.

	$r_{1}$	$r_{2}$	$r_{3}$	$r_{4}$	$r_{5}$	$r_{6}$	$r_{7}$	$r_{8}$	$r_{9}$	$r_{10}$	$r_{11}$	$r_{12}$	$r_{13}$	$r_{14}$	$r_{15}$	$r_{16}$	$r_{17}$	$r_{18}$	$r_{19}$
$\delta_{S}$	0	1	1	1	0	0	0	0	0	0	0	0	0	0	0	0	1	0	0
$M_{S}$	0	0	1	0	0	0	0	0	0	0	0	0	0	0	0	0	0	0	1
$\dot{\psi }$	0	1	0	1	0	1	0	0	0	0	0	0	0	0	0	0	1	0	0
$v_{rl}$	0	1	0	1	0	1	1	1	0	1	1	1	0	1	1	0	1	0	0
$v_{rr}$	0	1	0	1	0	1	1	1	0	1	1	1	0	1	1	0	1	0	0
$F_{TieRod}$	0	0	0	0	0	0	0	0	0	0	0	0	0	0	0	0	0	0	1

**Table 2 sensors-23-00887-t002:** Fault–symptom relationships of $r_{1}$ to $r_{8}$ for condition monitoring of actuators of the steering system (case a).

	$r_{1}$	$r_{2}$	$r_{3}$	$r_{4}$	$r_{5}$	$r_{6}$	$r_{7}$	$r_{8}$
SteeringMotor1	1	1	1	1	1	1	1	1
SteeringMotor2	1	1	1	1	1	1	1	1

**Table 3 sensors-23-00887-t003:** Fault–symptom relationships $r_{1}$ to $r_{8}$ for condition monitoring of actuators of the driving system (case b).

	$r_{1}$	$r_{2}$	$r_{3}$	$r_{4}$	$r_{5}$	$r_{6}$	$r_{7}$	$r_{8}$
DrivingMotorleft	0	0	0	0	0	1	0	1
DrivingMotorright	0	0	0	0	0	1	0	1

## Data Availability

Data sharing not applicable.

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
