# Peer review of "Model-Based Condition Monitoring of the Sensors and Actuators of an Electric and Automated Vehicle"

_sensors, 2023, doi:10.3390/s23020887_

Round 1

Reviewer 1 Report

The paper investigates the model-based condition monitoring (MBCM) of the sensors an actuators of an electric and automated vechile, an essential measure to achive safe and economic autonomous driving. The authors' MBCM model finds faulty sensors and it is able to recofigure the sensor signal. In addition, the MBCM model detects faulty behaviour of the actuators with the reconfigured sensor signals. The analysis is applied to demonstrator vechile of the SmartLoad project. The core of the detection method is based on generation of the residuals as the symptoms through the theoretical model of the vehicle. The paper is clear and nicely written.  

The authors are to be congratulated for their excellent contribution to automotive engineering practice. I recommend the paper to be published in Sensors – very nice work indeed.

Author Response

Thank you very much for spending the time to read my work and comment on my work. It has given me great help and is also the motivation for me to continue my research.

Reviewer 2 Report

In this paper, to improve the reliability of the sensors and actuators,the author  developed model-based method for condition monitoring of the sensors and actuators in an electric vehicle.The author's study of constant monitoring of driving conditions  is substantial and interesting. The auther proves that the method is effective by simulation. I'd like to recommend the paper.However, the following comments need to get addressed:

1. The introduction needs to be refined.the auther should add some relevant references.

2. It is suggested to connect the state of the art to the paper goals in the introduction.

3. There are some errors in Figure 4 and Figure 7. It is recommended to check the comments of the individual figures.

4. The author introduced a series of references when discussing the related work of the paper, but did not compare with the relevant methods in the simulation experiment.

In general, I propose to accept the paper after minor revision.

Author Response

Please see the attatchment. 

Reviewer 3 Report

 The article entitled "Model-based condition monitoring of the sensors and actuators of an electric and automated vehicle" evaluates a method to identify steering system fails from readings of sensors. The authors simulated numerically different failure situations and correlated with predicting models based on the Kalman filter to identify which component was failed.

Writing & Presentation

 There are formatting mistakes in the text, which requires a review. Some are listed with additional comments as the following:

·         Change on page 2, line 81, “accelerator” to “accelerometer”;

·         The text misses definitions of equation variables like “tr”, “lf”, “lr”, Greek letter: beta and psi, rdyn etc. A complete review of the text is required;

·         Review equations 24 and 25, there are equals. The same problem with equations 30 and 31;

·      Do the authors refer “central axis speed” the velocity of wheels?

·         Page 9, line 241, and from page 15, line 375, 381, 382 and 391, change “chapter” to “topic”;

·         Line 441, change “above” to “below”;

·      Just as a warning, the citation of the reference [21] came before [20] in the text because of the position of figure 2.

 Language proofreading is recommended.

Technical content

 For a better understanding of the mathematical formulation, the authors may provide a free body diagram that generates the equations 16 and 17. What are the angles beta and psi? The text presents variables before the explanation or does not describe the meaning of these variables.

 What is the failure case for Table 2? As the text is very long, it is difficult to understand what is considered as failure cases and when the authors are changing from one failure to another. I suggest naming each fail case as, for example, “case a”, “case b” etc., and dividing into subtopics.

 The authors may describe what is the failure by “total loss fault”, “positive/ negative offset” and “outlier”. And how these fault types are quantified as zero or one for the classification.

 The results from figures 17 and 18 may present some sample graphics with calculated residues, pointing out where the detection failed. In which condition did the method fail in detecting the sensor fail? When and where the simulation starts the sensor fail from test circuit map? The results should describe the conditions of the simulation and the correlation between them.

Reviewer 4 Report

1. Section 3, the system process and principle of the system model were not explained clearly. Only the sub model of six sensors is given.

2. There is no introduction of the workflow of the system model in this paper.

3. Section 3.7, the introduction to the CarMaker platform was not clear enough. Whether the working conditions are the same while the simulated data collected from the CarMaker. It lacks of necessary explanation on whether the simulation data is reliable.

4. In this paper, a series of simulation work has been done. But it lacks of verification of experimental data。This is where improvement is needed, if possible.

Round 2

Reviewer 3 Report

Authors have answered, in the article and comments, to my queries satisfactory. They added the missing information and corrected the formatting mistakes, thus I think that the article can be considered for publication.